# Intragenomic Variability of ITS Sequences in *Bjerkandera adusta*

**DOI:** 10.3390/jof8070654

**Published:** 2022-06-22

**Authors:** Peter Pristas, Terezia Beck, Maria Piknova, Svetlana Gaperova, Martin Sebesta, Jan Gaper

**Affiliations:** 1Institute of Biology and Ecology, Pavol Josef Safarik University in Kosice, Srobarova 2, 04101 Kosice, Slovakia; peter.pristas@upjs.sk (P.P.); maria.piknova@upjs.sk (M.P.); 2Centre of Biosciences, Institute of Animal Physiology, Slovak Academy of Sciences, Soltesovej 4–6, 04001 Kosice, Slovakia; 3Department of Biology and Ecology, Faculty of Natural Sciences, Matej Bel University, Tajovskeho 40, 97401 Banska Bystrica, Slovakia; svetlana.gaperova@umb.sk; 4T-MAPY, Dvojkrizna 49, 82106 Bratislava, Slovakia; martin.sebesta9@gmail.com; 5Department of Biology and General Ecology, Faculty of Ecology and Environmental Sciences, Technical University, T. G. Masaryka 24, 96053 Zvolen, Slovakia; jan.gaper@tuzvo.sk

**Keywords:** *Bjerkandera adusta*, polypores, ITS, intragenomic variability, sympatry

## Abstract

*Bjerkandera adusta* is a species of common white rot polyporoid fungi found worldwide. Despite playing an important role in deadwood decay, the species strains are used in bioremediation due to its ability to degrade polycyclic hydrocarbons and some of them are important etiological agents of chronic coughs and are associated with lung inflammations. In our experiments, diversity within the species was investigated using molecular approaches and we found that sequence diversity seen at ITS sequence level is not due to cryptic speciation but to intragenomic variability of ITS sequences in this species.

## 1. Introduction

*Bjerkandera adusta* (Willd.) P. Karst. (Agaricomycotina, Basidiomycota) is a cosmopolitan, common white-rot polypore that grows on dead broadleaved wood and damaged places of living trees and occurs on many woody plant hosts [1,2,3]. Morphological and phylogenetic studies have traditionally accepted the species in the genus, together with *B. fumosa*, both described from Europe [4]. However, recently, several new species within the genus were described, e.g., *B. albocinerea* [4], *B. atroalba*, *B. centroamericana* [5] and *B. mikrofumosa* [6] from the Neotropics.

*B. adusta* perfect stage (teleomorph) is characterized by pileate, effused-reflexed to resupinate basidiomata, cream to buff, then greyish to greyish-blue pileal surface, round to angular pore shape, tiny pores (6–7 per mm), grey to black pore surface, and short-cylindrical to subellipsoid basidiospores measuring 4.5–6 × 2.5–3.5 μm [1].

Moreover, few cases of isolating the *Geotrichum*-like imperfect stage of *B. adusta* lacking basidiospores and producing asexual arthroconidia (anamorph) were reported, namely from the fungal culture on agar plate with preferred medium [7,8], the Belize biodeteriorated compact disc [9], cultivated soil using pulp after industrial production of daunomycin as a substrate in Poland [10,11], and from Asian sand dust aerosol [12]. The *Geotrichum*-like imperfect stage of *B. adusta* isolated from this latter substrate induces allergic lung diseases.

This fungus has a symptomless endophytic life stage, within the roots of common reed *Phragmites australis* from Germany [13], the thallus of moss *Sphagnum fuscum* from Canada [14], xylem of Chilean trees *Drimys winteri* [15], the lycophyte *Huperzia serrata*, needles and twigs of *Abies beshanzuensis*, the roots of *Sinosenecio oldhamianus* from China [16,17,18], leaves of *Sarracenia purpurea* from the USA [19], and within sapwood and leaves of *Hevea* spp. from Brazil, Peru and Mexico [20]. Endophyte sequences identified as *B. adusta* had a similar pattern, in which these sequences clustered together [20].

*B. adusta* is an efficient lignin degrader and it is also able to oxidize xenobiotic compounds including some environmental pollutants making it a potential candidate for bio-technological and environmental applications, such as decolorization of commercially used diazo, anthraquinone, triphenylmetane, azo, phthalocyanine and other dyes in textile, leather, cosmetic, pharmaceutical and paper industries [18,21,22,23,24]. In addition to attractive enzyme activities, *B. adusta* is capable of producing a wide range of metabolites with interesting biotechnological applications, e.g., disaccharide α-α-trehalose, or antimicrobial activities [21,25].

With the recent development of molecular techniques, cryptic diversity within numerous polypore morphospecies was recognized. Despite the abovementioned biotechnological and environmental importance of the *B. adusta*, there is only limited data on variability of the species. The morphological variability observed in *B. adusta* concerns the phenotypic variation of basidiomata, mode of growth, their organization and the degree of tubes development. Several forms (e.g., f. *resupinata*, f. *solubilis*, f. *tegumentosa*) were described during the species history [2,26]. However, molecular analysis did not reveal substantial variability (less than 0.55%) within ITS sequences of Korean *B. adusta* specimens [27].

The aim of the present study was to analyze the intraspecific genetic variability of *Bjerkandera adusta* using molecular approaches based on ITS and *tef1-α* sequences.

## 2. Materials and Methods

To analyze the variability in *B. adusta* specimens from Slovakia (central Europe), multiple basidiomata (18) were collected (see Appendix A). For total genomic DNA isolation, fresh basidiocarps were ground by oscillating mill (MM200, Retsch GmbH, Haan, Germany) and DNA was isolated from a small amount of basidiocarp tissue (about 100 mg) using E.Z.N.A. Fungal DNA Mini Kit (Omega Bio-tek, Inc., Norcross, GA, USA). Quality and quantity of isolated DNA was analyzed using electrophoresis in 1.5% agarose gel.

The ITS and *tef1-α* regions were amplified using primer pair ITS1 (5’-TCCGTAGGTGAACCTGCGG-3’) and ITS4 (5’-TCCTCCGCTTATTGATATGC-3’) for ITS region amplification [28] and primer pair EF595F (5’-CGTGACTTCATCAAGAACATG-3’) and EF1160R (5’-CCGATCTTGTAGACGTCCTG-3’) for *tef1-α* amplification using conditions specified in Naplavova et al. [29]. The amplification products purified using ExoSAP-IT (Affymetrix, Inc., Cleveland, OH, USA) and sequenced in both directions using the same primers as for PCR at SEQme s.r.o. (Dobříš, Czech Republic). The chromatograms were manually checked for polymorphisms at specific nucleotide positions.

The sequences were checked for similarity against the GenBank database using blastn algorithm and deposited into the database under accession numbers ON391760–ON391777 for ITS sequences and ON411223–ON411229 for *tef1-α* sequences. For identification of polymorphisms, multiple sequence comparisons using MEGA-X software were used [30].

For RFLP analysis, approximately 0.5 µg of amplified ITS amplicons were cleaved with 5U of BstUI restriction endonuclease (Thermo Fisher Scientific, Inc., Cleveland, OH, USA) at 37 °C in R buffer as specified by the manufacturer. Restriction fragments were analyzed using electrophoresis in 1.5% agarose gel and visualized by Molecular Imager, ChemiDocTM XRS+ system (Bio-Rad Laboratories, Inc., Hercules, CA, USA) using ethidium bromide (0.5 µg/mL).

## 3. Results and Discussion

The internal transcribed spacer (ITS) sequence analysis is generally recognized and widely used for identification, systematics and phylogenetics of fungi. Sequence comparisons of *B. adusta* ITS sequences available in GenBank database (436 sequences by the end of March 2022) indicated the existence of two type of sequences in the analyzed dataset. Two types of sequences (referred to in this paper as type A and type B) differ by five transitions. One of the transitions is located in the ITS1 region, the rest of the transitions are located in the ITS2 region (Figure 1). In general, the ITS1 region is considered to be more variable than the ITS2 region within the Fungi kingdom; this variation included length and GC content variations as well as polymorphisms [31]. Our comparisons indicate that no clear geographical pattern could be recognized among type A and B sequences, as both types of sequences were reported worldwide. Analysis of available *B. adusta* ITS sequences (436 GenBank entries) indicated that a weak geographical pattern could be recognized among type A and B sequences. Although both types of sequences were reported worldwide, it seems that type B sequences are more frequently detected in specimens from Europe. For example, no type B sequences were found among 72 entries from the USA and only 2 from 206 entries from China were classified as type B. On the other hand, 13 out of 48 entries from Europe possess GA sequence in 467–468 position in the ITS2 region and can be classified as type B (data not shown).

Sequence comparisons of Slovakian *B. adusta* specimens showed that 7 out of 18 obtained sequences could be classified into type A and 3 into type B sequences. The remaining 8 sequences could not be typed due to the ambiguity of the sequences, observed at all polymorphic positions (marked as AxB type, Appendix A). Careful examination of GenBank data showed that similar ambiguity can be seen in several GenBank entries as well. Similar situations as seen in our study have been detected in other research, e.g., in *B. adusta* sequences from Chile. From the 18 sequences available, 3 can be classified as type B, 13 as type A and the remaining 2 sequences show the same polymorphism as detected in our specimens. 

A closer inspection of sequencing chromatograms revealed that the ambiguities occurred in polymorphic but no other sites, and rather than sequencing errors, these basidiomata might possess both types of ITS region sequences (Figure 2).

To evaluate the polymorphism at nucleotide position 522 in ITS sequences of *B. adusta* (G→A transition), a RFLP method was developed. The G→A transition at this position led to the loss or appearance of CGCG tetranucleotide, a recognition site for BstUI restriction endonuclease. RFLP analysis confirmed that both types of ITS sequences are present simultaneously in multiple *B. adusta* basidiomata (Figure 3) and no profiles other than the expected restriction profiles were observed. Cleavage by BstUI restriction endonuclease yielded a restriction fragment with size of about 400 bp and two smaller fragments of about 125 and 85 bp in type A isolates. Due to the transition at nucleotide 522 in type B isolates and consequent loss of BstUI site, these specimens produced a banding pattern possessing just two fragments of about 500 bp and 125 bp. In specimens possessing both types of ITS sequences, both 400 and 500 bp fragments were seen simultaneously. Comparison of the intensity of bands indicated that in specimens possessing both types of ITS sequences, type B clearly dominated over type A ITS sequences. 

Based on the data, it could be concluded that more than half of the basidiomata tested (either by ITS sequencing or RFLP analysis) possessed two different copies of ITS sequences differing by 5 nucleotides (0.9%) in their genome. 

The level of sequence polymorphism observed in *B. adusta* ITS sequences is much higher than the average error rates of Taq polymerase (between 0.1% and less than 0.01%, e.g., Potapov and Ong [32]) and is very close to the threshold values for delimiting fungal species. Some closely related sister species show similarities at ITS level as high as 99.5% [33,34]. Generally, similarity level 98.5% is used for species delineation, e.g., by UNITE and ISHAM databases [35,36]. The presence of intragenomic variability within ITS sequences could hamper correct identification of fungal specimens, especially if ITS amplicons are cloned in *Escherichia coli* vector prior to sequencing.

Intragenomic variability within ITS sequences of *B. adusta* was not accompanied by variability at other markers widely used for species delineation in fungi, the partial translation elongation factor (*tef1-α*) sequence. Partial *tef1-α* sequences (510 nt) were amplified from both ITS type A and B basidiomata as well as from specimens having hybrid ITS sequences. However, all obtained *tef1-α* sequences were completely identical, irrespective of ITS type (data not shown). Generally, *tef1-α* sequences show a higher degree of divergence in closely related species compared to ITS sequences [29,37,38] and the lack of variability seen in *B. adusta* probably indicates the genetic homogeneity of the species. Based on these data, we conclude that sequence diversity seen at ITS sequence level in *B. adusta* is not due to cryptic speciation but due to intragenomic variability of ITS sequences in this species.

Intragenomic ITS variability is a well-known phenomenon for bacteria, plants and animals but there is limited evidence on the occurrence of such variability in fungi (e.g., Hughes et al. [39], Stadler et al. [40]), mainly in Ascomycota. In a well-documented example of intragenomic ITS variability in Basidiomycota, Hughes et al. [39] observed variability at 28 ITS positions in *Amanita* cf. *lavendula* collections in eastern North America, Mexico and Costa Rica. In the smut fungi *Ceraceosorus* spp. [41], intragenomic variation of ITS sequences varied in up to four sites only, similar to the extent of ITS variability seen in our experiments. Similarly, in a polypore fungus *Trichaptum abietinum*, three nonorthologous ITS1 types were detected [42] and Type I and Type II ITS1 sequences were found to coexist in all tested *T. abietinum* strains. 

There is no clear explanation for the observed variability of ITS sequences. Possibly, observed variability could represent an initial state of hybridization event between two divergent taxa as proposed by McTaggart and Aime [43]. However, due to a limited number of specimens possessing pure type B ITS sequences, we were unable to identify some geographical, ecological or substrate specialization differences separating this group from type A group. In our collection, both type A and type B specimens were found within less than 1 km distance in Bratislava municipality and similarly, both type A and type AxB specimens were found in Sala municipality. Likewise, the examination of data from GenBank database indicated that both type A and type B specimens can occur sympatrically.

## Figures and Tables

**Figure 1 jof-08-00654-f001:**
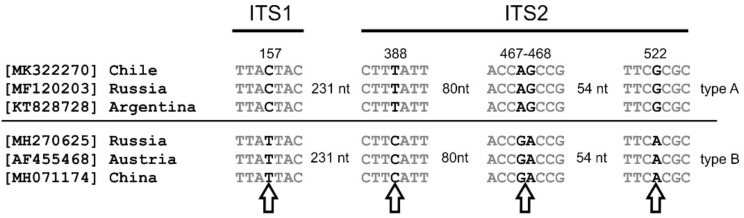
Variability of ITS region in *Bjerkandera adusta* based on alignment of sequences available in GenBank database. Polymorphic sites (marked by arrow) are shown in bold, together with nucleotide numbering according to the *B. adusta* strain RGM157 ITS sequence (GenBank accession number MK322270).

**Figure 2 jof-08-00654-f002:**
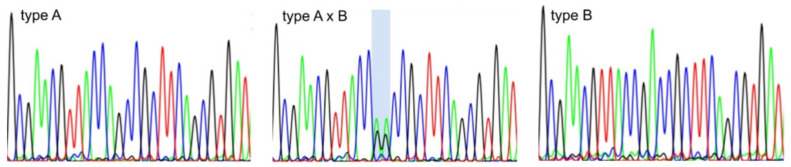
Comparison of parts of abi chromatograms showing polymorphism of ITS sequences in *Bjerkandera adusta*. The presence of both types of ITS sequence are shown on a blue background.

**Figure 3 jof-08-00654-f003:**
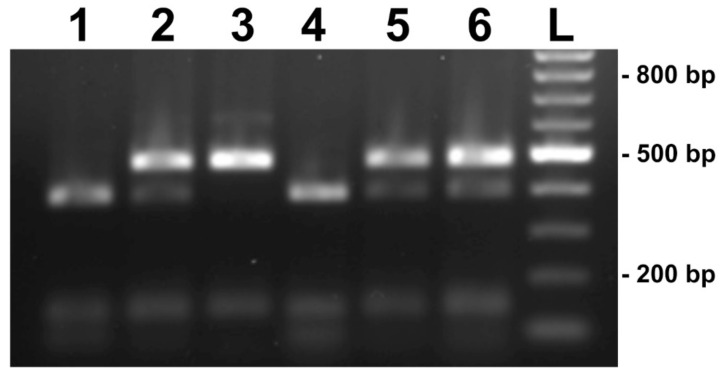
BstUI restriction fragment length polymorphisms of PCR amplified *Bjerkandera adusta* ITS regions. Lane1, *B. adusta* B1; Lane 2, *B. adusta* 859; Lane 3, *B. adusta* B2, Lane 4, *B. adusta* 692; Lane 5, *B. adusta* 884; Lane 6, *B. adusta* 994. L—standard of molecular weight. Gel electrophoresis in 1% agarose.

## Data Availability

All sequence data are available in NCBI GenBank following the accession numbers in the manuscript.

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
