# Peer review of "Intragenomic Variability of ITS Sequences in Bjerkandera adusta"

_jof, 2022, doi:10.3390/jof8070654_

Round 1
Reviewer 1 Report
The intragenomic ITS variability in Bjerkandera adusta from Slovakia has been investigate. Results showed the presence of two types of sequences in the same geographical area, as well as in northern hemisphere.
The manus may be appropriate for a communication paper, but not for a research paper, according to the authors' submission. I only have a few minor corrections to suggest, which are listed below:
Fix Bjerkandera in the title, instead of Bjerkander
Line 19: Delete the second ‘due’
Lines 26-27: In reference n. 4 the authors conclude that “the number of taxa in Bjerkandera has been underestimated by morphological evidence, and may actually be greater than traditionally accepted.” I suggest including information about the possible presence of additional species from countries other than Europe.
Line 37: add the reference n. 10 after ‘sand dust aerosol’
Line 38: better ‘isolated from this latter substrate’
Line 57: The epithets resupinata, solubilis, and tegumentosa should be written in italics
Line 63: please provide Table S1 with a detailed legend.
Lines 81, 95, 103, 111, 115, 139… and so on: B. adusta in italics
Lines 95-96: you can delete this sentence, that is the same of lines 62-63
Line 138: Escherichia coli in italics
Line 154, 155, 157, and 159 Amanita, lavendula, Ceraceosorus, and Trichaptum abietinum in italics
Author Response
Dear Reviewer 1,
we accepted all your comments and suggestions and therefore we hope that communication is acceptable for publication in JoF.

Reviewer 2 Report
Intragenomic variability of ITS sequences in Bjerkandera adusta
By Peter Pristas, Terezia Beck, Maria Piknova, Svetlana Gaperova, Martin Sebesta, Jan Gaper
Some suggestions are provided to improve the manuscript:
2 – Species name is incomplete Bjerkander adusta= Bjerkandera adusta.
40 – There are many reports of B. adusta on different hosts, and authors reports as an endophyte, but I remember several records in commercial forest of Eucalyptus spp. in South America, both, as endophyte and associated on stumps that deserves interest.
47 – There are hundreds of works on applications of B. adusta in biotechnology, like some cited here, and this makes it interesting. But it would be better to highlight here the most important aspects, and not just particular ones. Not only peroxidases and trehalose are important.
51 – 52 - With the recent development of molecular techniques, … to another paragraph.
51 – 60 – This part is a bit messy, and the objectives need to be clearly outlined.
75 – “The sequences were checked against GenBank database” … Only this? Sequences were not aligned and managed with a particular software?
77 – It is not clear what was done. The sequences were compared/checked by means of Blastn only? Were all the sequences from Slovakia (18) and the 437 deposited in Genbank compared?
79 – Why weren't all sequences aligned and compared by phylogenetic methods?
81 – B. adusta in italics. Correct throughout the text (e. g. lines 95, 96, 103, 139, 146, …) as well as other scientific names.
90 – There is no correlation with geography, but here it seems that type A of ITS sequences appears in Russia and South America, and type B in Eurasia. Why did you choose these sequences to compare/show? Why not from different/more continents?
95 – It seems better to present these results, compared to international sequences, than to show only the sequences shown in Figure 1 as example.
105 – A closer inspection of sequencing chromatograms … this is not indicated in Materials and Methods.
112 – RFLP method was developed… how it was made?
124 – In Figure 3, strains/specimens (as presented in Supplementary Material) are not showed associated to each band pattern.
125 – Also, the RFLP is not mentioned in Materials and Methods section.
163 – Only 3 sequences with Type B ITS were found in the 437 reported sequences in Genbank?
Author Response
Dear Reviewer 2,
we accepted most of your comments and suggestions and therefore we hope that communication is acceptable for publication in JoF. Please see the attachment.
Yours sincerely, Team of authors

Round 2
Reviewer 2 Report
Comments and Suggestions for Authors on manuscript jof-1767609
Intragenomic variability of ITS sequences in Bjerkandera adusta
By Peter Pristas, Terezia Beck, Maria Piknova, Svetlana Gaperova, Martin Sebesta, Jan Gaper
Thank you for the answers provided and the corrections made on the suggestions provided in the first revision.
At this point I think the communication is ready to be accepted for publication after a minor correction in the text and English.
A minor suggestion/correction:
38 – 39 – “the Belize biodeteriorated compact disc CD” … it would be better “a biodeteriorated compact disc”, CD is redundant.
Author Response
Dear Reviewer 2,
We accepted all your comments and therefore we hope that communication is acceptable for publication in JoF.